# Effects of Weak Acids on the Microbiological, Nutritional and Sensory Quality of Baltic Herring (*Clupea harengus membras*)

**DOI:** 10.3390/foods11121717

**Published:** 2022-06-12

**Authors:** Nora Logrén, Jaakko Hiidenhovi, Tanja Kakko, Anna-Liisa Välimaa, Sari Mäkinen, Nanna Rintala, Pirjo Mattila, Baoru Yang, Anu Hopia

**Affiliations:** 1Functional Foods Forum, Faculty of Medicine, University of Turku, 20014 Turku, Finland; nanna.rintala@utu.fi (N.R.); anuhop@utu.fi (A.H.); 2Food Processing and Quality, Production Systems, Natural Resources Institute Finland (Luke), Myllytie 1, 31600 Jokioinen, Finland; jaakko.hiidenhovi@luke.fi (J.H.); sari.makinen@luke.fi (S.M.); 3Food Chemistry and Food Development, Department of Life Technologies, University of Turku, 20014 Turku, Finland; tatese@utu.fi (T.K.); bayang@utu.fi (B.Y.); 4Food Processing and Quality, Production Systems, Natural Resources Institute Finland (Luke), Paavo Havaksen tie 3, 90570 Oulu, Finland; anna-liisa.valimaa@luke.fi; 5Food Processing and Quality, Production Systems, Natural Resources Institute Finland (Luke), Itäinen Pitkäkatu 4, 20520 Turku, Finland; mattilapirjoh@gmail.com

**Keywords:** pickling, fish, sensory profile, organic acids, chemical composition, lipid oxidation, microbiological safety

## Abstract

Baltic herring (*Clupea harengus membras*) pickled in vinegar is a common product in the Nordic countries. Other weak acids are used to cook and preserve fish in other food cultures. The aim of this study was to evaluate the potential of weak acids to produce safe and nutritious pickled fish products with varying sensory properties. The influence of acetic, citric, lactic, malic, and tartaric acids on the preservability and quality of pickled and marinated Baltic herring was studied by measuring microbiological quality, pH, chemical composition, and lipid oxidation and by sensory profiling. Pickling with these acids with pH levels of 3.7–4.2 resulted in pickled Baltic herring products with high microbiological quality. The results of the chemical analysis of the samples indicated that pickling and storage on marinade influenced the chemical composition of fish. The most significant changes in chemical composition were the increase in moisture and decrease in protein content of the samples during storage. Fat content decreased during the storage period in acetic acid and malic acid samples. All tested acids inhibited lipid oxidation for one month, but at three and four month time points, the content of oxidation products increased except in the samples pickled with tartaric acid. The highest oxidation level was observed in the case of citric acid and the lowest with tartaric acid. The results indicate that replacing acetic acid with other weak acids frequently used in the food industry results in pickled and marinated fish products with novel and milder sensory profiles.

## 1. Introduction

Globally, acid treatment is a widely applied method to both cook and preserve different types of fish products. Pickling with the help of acid and salt is among the oldest preservation techniques for fish. The acid and salt treatment tenderizes the fish muscles in a similar manner to heat and inhibits the growth of spoiling bacteria [1]. In Europe, the most common acid used for producing pickled fish is acetic acid. Vinegar-based fish pickled with various herbs, spices and other seasonings represent a wide range of products. The preparation of canned Baltic herring (*Clupea harengus membras)* in Finland is based on pickling the herring in acetic acid. Although vinegar is the most common added acidic pickling agent, acidity can also be achieved through the production of acid by fermenting acetic and lactic acid bacteria. Lemon and lime juice rich in citric acid are used in many traditional fish recipes such as in the originally Peruvian dish, ceviche. In fermented fish products, such as Swedish surströmming, lactic acid produced by lactic acid bacteria is the source of acidity.

Under acidic conditions, the growth of food-borne pathogens and food spoilage microbes is inhibited. *Listeria monocytogenes* is a food-borne pathogen that might cause listeriosis, a disease that can even lead to death. [2]. The bacterium is ubiquitous in the environment, with ready-to-eat (RTE) fish and fishery products being a particularly potential vehicle for the microbes [3]. The growth of *L. monocytogenes* may vary depending on the strain and matrix. In a broth media, the lower growth limit is at pH 4.0–4.3 [4]. Organic acids have been widely used in the food industry due to their antimicrobial nature [5]. 

In addition to inhibition of bacterial growth, a low pH also changes the muscle structure as protein denaturation is dependent on the pH. Protein is most prone to denaturation at a pH below or above the isoelectric point due to the increase in the net charge of the protein, inducing protein-protein repulsion, and unfolding and swelling. The swelling of the muscle tissue caused by a low pH is likely to increase tenderness [6,7]. At the isoelectric point, protein is most resistant to denaturation [8]. Most previous studies on the acid treatment of muscle foods using weak organic acids have only included comparisons of different concentrations of the same acid [9,10,11], and/or the same concentrations of different acids [9]; however, few studies have addressed the effect of different acids at a constant pH, despite the significance of pH in the process.

The pickling and marinating process affects the texture of meat via the protein degrading enzyme activity. It has been shown that during the marinating process the amount of nitrogen containing compounds decreases in Baltic herring fillets and carcasses, and primarily increases in the marinating solution [12]. This indicates that there is some proteolytic activity in the solutions during the pickling and marinating processes. The proteolysis catalyzed by, e.g., cathepsin D may increase the tenderness [7]. Cathepsins, found in lysosomes [8], are protein hydrolyzing proteases that break proteins down into smaller molecules of nitrogen containing compounds such as amino acids. Cathepsin D, an aspartic peptidase, is one of the most abundant cathepsin in lysosomes. It is stable at pH 3–5 [13,14] and most active at pH 2–4 [15] and it has been found to be in an active form in brine after marinating frozen-thawed Atlantic herring (*Clupea harengus*) for 7 days with a 5% acetic acid [16]. Burke and Monahan [9] suggest that collagen solubilization also contributes to tenderization after the marinating process and that there are two mechanisms by which collagen molecules may solubilize. The first mechanism suggested is peptide bond hydrolysis which increases as the pH decreases, and, thus, the collagen solubilizes. The second mechanism suggests that collagen solubilizes through a breakdown of the cross-links that are sensitive to pH, heat, or denaturing agents.

Dark-muscled fish such as Baltic herring are prone to oxidation due to the high content of heme proteins, known to be significant endogenous pro-oxidants in fish [17]. Baltic herring is high in polyunsaturated fatty acids (PUFA), with as high as approximately one third of all the fatty acids being polyunsaturated [18,19]. Decreasing the pH of fish can increase lipid oxidation due to the deoxygenation of hemoglobin via the Root and Bohr effects, since deoxyhemoglobin is considered a better catalyst of lipid oxidation than oxyhemoglobin [17]. Moreover, the treatment of fish with organic acids has even been seen to produce a protective effect regarding oxidation [20,21].

Fish products pickled with vinegar have a characteristic flavor profile in which acidity and vinegary flavor dominate. The pickling process has an effect on the formation of taste and flavor compounds in the fish. Free amino acids produced by proteolysis may have sweet, sour, bitter, and umami taste properties [22] that can affect the flavor properties of the pickled and marinated herring products. Furthermore, the lipid oxidation products of unsaturated fatty acids in fish contribute to the flavor. However, the research regarding the sensorial quality of pickled Baltic herring products is limited. The topic has been studied with, e.g., acceptance assessment [11,23], hedonic evaluation [24,25] and by using the “Just about right–scale” [25]. The sensory properties of traditionally pickled and marinated Atlantic and Baltic herring have been studied by sensory profiling [26,27]. The traditional flavor profile is preferred by certain consumer groups. However, the consumption of Baltic herring has decreased by one third in the last 40 years. In order to obtain new consumers for pickled Baltic herring, products with milder sensory properties might be attractive.

To summarize, the chemical, microbiological and sensory studies on the pickling of fish have mostly been focused on pickling with acetic acid and the potential of other weak acids commonly used in food production has not been systematically documented for fish. The aim of this study was to evaluate the potential of food grade weak acids other than the commonly used acetic acid in order to develop pickled Baltic herring products with novel sensory profiles. The acids studied were citric, lactic, malic, and tartaric acid, acetic acid being the reference. The quality and preservability of the samples were studied by measuring the pH, composition, lipid oxidation, and microbiological quality and by sensory profiling.

## 2. Materials and Methods

### 2.1. Raw Material and Sample Preparation

The skinned Baltic herring fillets were provided by Martin Kala Ltd. (Turku, Finland). The fish were caught in the southern part of the Gulf of Bothnia on 27 October 2020 and the skinless fillets were delivered to Camilla’s Fiskdelikatess Ltd. (Molpe, Finland) within 2 days of being caught and kept fresh on ice. Food grade acids, citric acid monohydrate, malic acid, lactic acid, and tartaric acid (Vinoferm^®^, Brouwlan, Beverlo, Belgium), were purchased from Juomatarviketukku Lappo Ltd. (Piikkiö, Finland). Acetic acid (Berner Ltd., Helsinki, Finland) was provided by Camilla’s Fiskdelikatess Ltd.

The concentration and pH values of the acid solutions are presented in Table 1. The suitable concentration for acetic acid, citric acid, lactic acid, malic acid, and tartaric acid for the pickling and marinating solutions was determined by pre-tests, since the addition of fish increased the pH. This is due to the different strengths and buffering capacities (ability to maintain acidic pH after addition of the fish) of the organic acids. The aim was to achieve a pH of <4 at the end of pickling. The target of pH < 4 was chosen since it is a typical pH for fish preservatives treated with acetic acid, and is considered a safe pH with regard to limiting the growth of food pathogens, including *L. monocytogenes* [5]. Sodium benzoate was also added to the marinating solutions as a preservative.

The samples were produced at Camilla’s Fiskdelikatess Ltd., whereas the training samples for the sensory evaluations were prepared in the test kitchen of the Functional Foods Forum, University of Turku (Turku, Finland) (ISO-8589:1988). The concentration of salt in the pickling solutions was 5.0% (*w*/*v*). The concentrations of sugar and sodium benzoate in the marinating solutions were from the commercial recipes of Camilla’s Fiskdelikatess Ltd. The fish-to-solution ratio in the pickling solutions of the training and actual samples was 1:1.4, and 1:1.5 (*w*/*v*), respectively, and in the marinating solutions of the training and actual samples it was 1:1.2, and 1:1.3 (*w*/*v*), respectively. The pickling process was conducted in 150 l plastic barrels commonly used in the manufacturing of pickled herring products. The pickling was conducted at 2–3 °C for 24 ± 3 h. The fillets were removed from the pickling solution and placed on a grating to drain the excess solution. The fillets were weighted down in glass jars with a grating to keep the herring under the marinade. The marinade was added, and the jars were vacuum packed. The samples were kept sealed at 3–6 °C until analysis. The samples were analyzed at 0, 1, 3, and 4 month time points after preparation except for the sensory profile, which was created and reviewed twice after 0, 1, and 4 months, respectively. The 0 month’s analysis was started after the fish had been pickled and placed in the marinating solutions. 

### 2.2. Microbiological Quality

The number of L. monocytogenes from the fresh samples (at the time point 0 month) was determined according to the qualitative method SFS-EN ISO 11290-1:2017 in an accredited laboratory SeiLab Ltd. (Seinäjoki, Finland). The horizontal method is used for the detection and enumeration of *L. monocytogenes* and of Listeria spp. in food and feed [28]. The result was reported as found/not found per 25 g of the sample. At the time points 1 month and 4 months the number of bacteria was analyzed by the qualitative method Vidas LMX (*) Vidas Express (LMX), ISO 11290-2:2017 in the accredited laboratory ScanLab Ltd. (Oulu, Finland). The result was reported as found/not found per 25 g of the sample.

The number of sulfite-reducing bacteria was determined according to the method of the Nordic Committee on Food Analysis NMKL 56:2015 NMKL Method No. 56, 5. Ed., 2015 in ScanLab Ltd. (Oulu, Finland). The result was reported as the number of viable anaerobic sulfite-reducing bacteria per gram of the samples, i.e., the number of colonies forming units per gram of sample (cfu/g). The results were expressed on a logarithmic scale.

The number of aerobic microorganisms at 30 °C were determined by the colony count method according to the NMKL Method No.86, 5th Ed., 2013 used by ScanLab Ltd. The result was reported as cfu/g expressed on a logarithmic scale.

The number of aerobic psychro-trophic microorganisms, the number of yeasts, and the number of hydrogen sulfide-producing bacteria were determined in the Natural Resources Institute Finland (Luke) (Oulu, Finland). For all these analyses, 10 g of sample was mixed with 90 mL of sterile Maximum Recovery Diluent (Lab M Ltd., Heywood, UK), and homogenized by a Bagmixer 400 (Interscience, Saint Nom la Bretêche, France) for 1 min. A ten-fold serial dilution was also prepared with the same media. After incubation on the appropriate agar under the microbial specific growth conditions the number of colonies on the agar plates was counted. The result was reported as cfu/g expressed on a logarithmic scale. 

The number of aerobic psychro-trophic microorganisms were determined by the colony counting method according to the NMKL Method No.86, 5th Ed., 2013 in the Natural Resources Institute Finland. Using this method, the dilutions of the samples are pipetted into Petri dishes, and Plate Count Agar (PCA) (Lab M Ltd., Heywood, UK) was poured into Petri dishes and the suspension was mixed. The plates were incubated under aerobic conditions at 6.5 °C ± 1.0 for 10 days.

The number of yeasts were measured by a dilution plating technique according to the Nordic Committee on Food Analysis NMKL Method No.98, 4th Ed.,2005 using oxytetracycline-glucose yeast extract agar (O.G.Y.E.) (Lab M Ltd., Heywood, UK). The dilutions of the samples were spread on the O.G.Y.E. agar surface in Petri dishes. The agar plates were incubated under aerobic conditions at 25.0 °C ± 1.0 for 5 days. 

The number of hydrogen sulfide-producing bacteria were measured by Iron Agar (Lyngby) (Oxoid Ltd., Hampshire, UK) with a 4% *w*/*v* solution of L-cysteine hydrochloride monohydrate (Sigma-Aldrich, St. Louis, MO, USA). The media was prepared according to the manufacturer’s instructions. The dilutions of the samples were spread on the Lyngby agar surface in Petri dishes. The plates with *Shewanella putrefaciens* ATCC 8071 produce black colonies and *Pseudomonas fluorescens* and ATCC 13525 which produces white colonies on the Lyngby agar was used as the reference microorganisms. The agar plates were incubated under anaerobic conditions at 25.0 °C ± 1.0 for 48 h. The black colonies, with characteristics that were comparable to the colonies of *S. putrefaciens* ATCC, 8071 were counted as hydrogen sulfide-producing bacteria. 

### 2.3. pH

The pH values in the solution of the canned samples were measured at 0, 1, 3, and 4 months. Measurements were conducted in duplicate from three different cans using a FiveEasy pH meter equipped with an LE409-DIN electrode (Mettler Toledo, Columbus, OH, USA).

### 2.4. Composition

The moisture and ash contents in the samples were determined by using a TGA701 Thermo-gravimetric Analyzer (Leco Corporation, St. Joseph, MI, USA). The TGA measured weight loss as a function of temperature (105 °C and 650 °C for moisture and ash, respectively) under controlled conditions, and automatically determined the moisture and ash contents in the samples with the selected program. The nitrogen content was determined with an in-house Kjeldahl method based on ISO 20483, ISO 5983-2 and AOAC 2011.11 methods by using a Kjeltec TM8400 analyzer (Foss Analytical Ltd., Höganäs, Sweden). A conversion factor of 6.25 was used to calculate total protein content. The total fat content was determined using the SoxCap TM 2047 in combination with the Soxtec TM 2050 extraction system (Foss Ltd., Hillerød, Denmark) with a preparatory acid hydrolysis step and diethyl ether extraction according to ISO 6492. The total carbohydrate content was calculated with the following formula:carbohydrates (% fresh weight, FW)  =  100 − moisture (%) − protein content (%FW) − crude fat (%FW) − ash (%FW).(1)

### 2.5. Lipid Oxidation

The effect of the different acids on lipid oxidation in the Baltic herring fillets was assessed by measuring the thio-barbituric acid reactive substances (TBARS) in the fish samples. The TBARS in the fish samples were measured after 0, 1, 2, and 4 months of storage. Prior to the analysis, the Baltic herring samples were subjected to alkaline hydrolysis which releases malondialdehyde (MDA) from proteins according to Mäkinen et al. [29]. Briefly, the samples were homogenized, and three 100 mg subsamples of each homogenized sample were taken for alkaline hydrolysis. The hydrolysis was conducted by mixing the subsamples with NaOH and incubating the suspensions in a 60 °C water bath for 30 min. After the hydrolysis, the proteins were precipitated with sulfuric acid and trichloroacetic acid (TCA), and supernatants were collected after centrifugation. The supernatants were then reacted with thio-barbituric acid (TBA) to form MDA-TBA adducts with a pink pigment. Samples were then analyzed with UHPLC at 532 nm. The concentration of MDA was calculated using the MDA-TBA standard curve (12.5–800 μM). Samples were analyzed in triplicates. Chromatographic analyses were performed in duplicate from each of the subsamples (*n* ≥ 6). Results are expressed as mean ± SD.

### 2.6. Sensory Profile

The sensory properties of the pickled and marinated Baltic herring samples were analyzed with a generic descriptive method. A panel (*n* = 10, 10 women) was recruited from a group of trained sensory panelists. The panelists had previous experience in sensory evaluation of various food samples, and the acuity of their senses had been evaluated. The panel was then specifically trained in analyzing marinated Baltic herring samples during four training sessions. The training of the panel at 0 month time point was done with samples prepared earlier but with the same procedure as the actual samples (Appendix A). Due to the prevailing COVID-19 pandemic situation the training sessions were held with only 3–4 people present per session except at the 4-month evaluation where only 2 panelists took part in the training simultaneously. A list of sensory attributes was created by the sensory panel to describe the characteristics of the pickled herring samples. Suitable reference samples were defined to reflect the corresponding sensory attributes and their intensities. The attributes were evaluated on a scale from 0–10 verbally anchored at the ends (0 = not at all, 10 = very strong). The attributes and reference samples are presented in Table 2.

Sensory evaluations were carried out during four sessions in the sensory laboratory of the Functional Foods Forum, University of Turku (ISO-8589:1988). The samples were coded with random three-digit numbers and offered at room temperature in randomized order. Compusense^®^ Cloud software (Version 21.0, Compusense Inc., Guelph, ON, Canada) was used for data collection. To confirm the safety of the products evaluated, the number of *L. monocytogenes*, the number of aerobic microorganisms and the number of sulfite-reducing bacteria (see 2.4. Microbiological quality) were determined before the time point of each sensory evaluation to ensure the microbiological safety of the samples.

The first sensory evaluation at 0 months was held on day 6 after the pickling of the herring. The 1- and 4-month sensory evaluations were carried out in a similar manner after five and 17–18 weeks of the sample preparation, respectively. For the 1- and 4-month evaluations, the same sensory panel was recruited, with nine of the ten panelists able to participate at 1 month and six at 4 months. An additional sensory evaluation was held to collect sufficient data at the 4-month evaluation. A training session was held before the subsequent sessions in order to recall and discuss the attributes before the evaluations. 

### 2.7. Statistical Analysis

Statistically significant differences between pH values were determined using IBM SPSS Statistics 25 software and the statistical analyses for the chemical composition, lipid oxidation, and sensory analyses were executed with IBM SPSS Statistics 27 (Armonk, NY, The United States). The limit for the statistical significance level was set at *p* < 0.05. Statistically significant differences between treatments were calculated with a one-way analysis of variance (ANOVA) test and Tukey’s honestly significant difference post-hoc test. Tamhane’s T2 post-hoc test was also used in the statistical analysis of the sensory analysis depending on the equivalence of variance.

Differences in pH values and sensory properties of each acid treatment between the time points were tested with paired-samples t-test and repeated ANOVA measures depending on the number of variables. Bonferroni and Šidák corrections were used to correct the *p*-values. A paired samples *t*-test was conducted to compare the results of the chemical analysis at different time points within the same acid treatment.

## 3. Results

### 3.1. Microbiological Quality

*L. monocytogenes* bacteria was not found in any samples tested. The detection limits were 1 cfu/25 g of the sample. The number of sulfite-reducing Clostridia in all the tested samples was below the detection limit of <10 cfu/g.

The number of aerobic microorganisms was generally very low in all the tested samples (Table 3). The highest count was observed in the fresh (0 months) citric acid (2.6 log cfu/g) followed by lactic acid (2.0 log cfu/g), malic acid (1.9 log cfu/g), and tartaric acid (1.5 log cfu/g) samples. The acetic acid sample had the lowest bacterial count (1.0 log cfu/g) of the samples tested. The number of aerobic microorganisms had already decreased after one month of storage and remained low during the 4-month storage period. The bacterial count was below the detection limit of <1.0 log cfu/g in the acetic acid and tartaric acid samples. In the citric acid sample, the bacterial count decreased by 1 log, and in the lactic acid and citric acid samples by 0.6 log.

The number of aerobic psychro-trophic microorganisms and the number of yeasts in all the tested samples were below the detection limit of <10 cfu/g. The number of hydrogen sulfide-producing bacteria in the fresh citric acid sample was 10 cfu/g. The number of bacteria in all the other samples tested was below the detection limit of <10 cfu/g.

### 3.2. pH

The pH values of the canned samples at the different timepoints are presented in Table 4. Despite the high pH of some of the marinade solutions (Table 1), a pH of <4 was mostly achieved in the canned samples. With most of the acids, the pH increased slightly over the first three months of storage.

### 3.3. Composition

Initially, the moisture content was a little over 50% in all samples and gradually increased during the 4-month storage to reach about 60% in the acetic acid and citric acid samples and 56–58% in the other samples (Figure 1).

The changes in protein content in Baltic herring fillets during the marinating process are shown in Figure 2. The initial protein content of the samples varied from 13.5% in the acetic acid sample to 14.7% in the tartaric acid sample. All the acids had a significant effect on the protein content from day 0 and this further increased during the storage in marinade. After the 4 months storage period the protein content levels decreased most significantly in the acetic acid sample, which went down to 7.8%; in comparison the protein content of the malic acid and tartaric acid samples remained highest, 12.9 and 12.6%, respectively.

The carbohydrate content of the samples was approximately 30% in the samples immediately after the sample preparation and decreased slightly during the storage period to 26–28% in all the samples (Figure 3). 

The fat content of the Baltic herring fillets samples on the different timepoints are presented in Figure 4. At the beginning of the storage period, the fat content of the fillets was approximately 3% and decreased during the storage in all the samples except the lactic acid sample. The natural variation in the fat analyses was high. In the acetic acid samples, the fat content decreased the most, from 3.2% to 1.4%. 

### 3.4. Lipid Oxidation

Fish fat is a highly specific nutritious element, due specifically to its high content of PUFAs. MDA has been widely used to represent the degree of lipid oxidation in various meat products. The production of MDA occurs in the second phase of lipid oxidation, during which peroxide is oxidized to aldehydes and ketones [30]. In the present study, we measured how the different acids used in the marinade affect the content of MDA in the Baltic herring fillets during four months of cold storage (Figure 5).

The initial concentration of MDA in the Baltic herring fillets immediately after preparation varied from 1.0 ± 0.1 μM/g dry matter (dm) to 1.3 ± 0.1 μM/g dm; the lowest content was detected in the fillets in the citric acid marinade and the highest value in the fillets in the acetic acid marinade (Figure 5). The MDA content remained at the same level until after 1 month of cold storage, with all tested acids showing no statistically significant differences between the 0- and 1-month time points. However, the MDA levels increased significantly in all the tested acids when the storage period continued to 3 and 4 months except for tartaric acid (Figure 5, *p* < 0.05). After 3 and 4 months of cold storage, the citric acid marinated fillets were evidently the most oxidized sample. The highest MDA level (2.1 ± 0.2 μM/g dm) was observed after 4 months of cold storage in Baltic herring filets marinated with citric acid. After the 4 months of cold storage, Baltic herring fillets marinated with citric acid showed a significantly higher MDA level in comparison to all the other tested acids (Figure 5, *p* < 0.05). After 3 months of cold storage, the MDA content in the Baltic herring fillets marinated with citric acid was significantly higher in comparison to the filets marinated with tartaric and malic acid (*p* < 0.05). After 3 and 4 months of cold storage, Baltic herring fillets marinated with tartaric acid showed the lowest MDA content, but the difference in comparison with the other acids was not statistically significant (*p* > 0.05).

### 3.5. Sensory Profile

The sensory panel evaluated the sensory properties of the five Baltic herring products at three time points at 0, 1, and 4 months. Their odor, texture, taste, and flavor properties were evaluated using 20 sensory attributes. The panel recognized seven odor attributes, two texture attributes, six taste attributes, and five flavor attributes. The odor attributes were total intensity, sweetness, vinegar, fish, stuffy, rancid, and metallic. Texture attributes were crumbliness and moistness. Taste attributes were total intensity, sweetness, saltiness, umami, sourness, and bitterness. Flavor attributes were vinegar, fish, stuffy, rancid, and metallic. The differences between sensory characteristics of the samples can be seen in Table 5. 

During the 4 months of preservation, there were statistically significant changes between the samples in 12 of the twenty sensory attributes: total intensity of odor and taste, vinegary and metallic odor and flavor, crumbly and moist texture, salty, sour, and bitter taste, and stuffy and metallic flavor. There were no differences between the samples at any point in eight of the sensory attributes: sweet, stuffy, and metallic odor, sweet, salty, umami, and bitter taste, and rancid flavor.

The acetic acid sample was clearly stronger than most of the other samples in total intensity of odor and taste, vinegary odor and flavor, sour taste, and crumbly texture. The differences were significant at all time points. The crumbly texture was more intense in the acetic acid sample than in the other samples after 0 and 1 months of storage. It was more intense in the acetic acid sample compared to the citric acid and malic acid samples after 4 months of storage. The intensity of the crumbly texture was above 9 in all samples except the malic acid sample after 4 months of storage. Although the fishy odor and flavor were recognized in all the samples, the acetic acid sample was noticeable as being the mildest in these attributes during the whole study. The tartaric acid sample was significantly stronger in total intensity of odor compared to the citric acid sample and in crumbliness compared to the malic acid sample after 1 month of storage. The tartaric acid sample was significantly stronger in metallic flavor compared to the malic acid sample after 4 months of storage.

The sensory quality changed during the preservation. The lactic acid sample had the greatest number of sensory attributes that changed during the 4-month storage period. In the lactic acid sample seven of twenty sensory attributes had statistically significant differences in intensities when comparing the sensory evaluations over the different months; metallic odor and flavor, crumbliness and moistness of texture, salty and bitter taste, and vinegary flavor and their intensity had increased over time. In contrast, in the malic acid sample the only sensory attribute that changed over time was its increasing crumbliness. The crumbly texture was the only attribute that changed in all the samples over time. 

## 4. Discussion

The influence of acetic, citric, lactic, malic, and tartaric acids on the preservability and quality of pickled and marinated herring was studied by measuring the microbiological quality, pH, composition, and lipid oxidation and by sensory profiling.

Organic acids are weak acids and vary in strength (pKa). For instance, acetic acid has a pKa of 4.76, making it a weaker acid than most other organic acids, such as tartaric acid (pKa = 2.98) or lactic acid (pKa = 3.86). The pKa is determined by the degree of dissociation in water and therefore the pH for a certain concentration of an acid [31]. However, different muscles vary in their buffering ability, i.e., their ability to resist the change of pH upon the addition of acid (or base) [32]. Due to the different buffering abilities of the acids, the acid concentrations in the pickling and marinating solutions varied as pH level in this study was aimed at 4. Since the differences between the samples regarding pH at the end of the pickling and marinating were so small, it seems likely that the acid accounts for the differences in the results of chemical analyses rather than the pH and different acid concentrations.

Food intended for human consumption must be microbiologically safe. According to the results the microbial quality of all the tested products was excellent in respect of both food-borne pathogens *L. monocytogenes* and the sulfite-reducing Clostridia and the food spoilage microbes including yeasts, aerobic mesophilic and psychro-trophic microorganisms and hydrogen sulfide-producing bacteria. This was expected due to the low pH (pH < 4.2) of the samples. In the weak organic acids used in our study, pH is considered the determining factor in their antimicrobial effects, since it is connected to the degree of dissociation. The suggested mechanism behind this effect is that, in an undissociated form, acids can pass through the bacterial cell wall, and in the neutral cytoplasm of the cell dissociate into H^+^ ions and anions, both of which can be detrimental to bacteria [31].

In addition to low pH, other environmental factors, including NaCl content, water activity, and storage temperature affect microbial growth. In this study, the effects of the aforementioned factors on microbial growth were not studied.

Pickling and storage in a marinade has an effect on the chemical composition of fish. In this study, the moisture content of the Baltic herring fillets increased, while both the protein and fat content decreased during the 4-month storage. This is in contrast to the results reported in previous studies [15,16,17]. This is probably due to two reasons: the low salinity of the pickling solution in the pickling procedure and the complete absence of salt in the marinade solution. The salinity of the water affects the solubility of the proteins. At low concentrations (<5.8%, here 5%) the proteins swell, and their solubility increases. [33,34,35] As a result, salt and water are absorbed into the muscle and soluble proteins and non-protein nitrogen compounds diffuse into the surrounding solution [12].

During the pickling and marinating processes, acid diffuses into the fish muscle, lowering the pH and causing protein denaturation and lower water absorption [17]. If the pH of the pickling solution is low and, thus, the pH of the muscle tissue is on the acidic side of the isoelectric point, it results in electrostatic repulsion of actin and myosin filaments, which causes an open structure in the muscle tissue and increases the water retention. However, when the salt content increases further, the repulsive charges are shielded, which lowers the water retention capacity. Therefore, salt is needed during the pickling and marinating process, as the use of too low salt content results in soft fillets [36]. In future studies and product development the optimal level of salt in the pickling and marinating solutions should be carefully adjusted.

Despite a direct comparison of the MDA levels of the marinated Baltic herring fillet with those in the literature being difficult, due to the different methods and units used, the levels measured in this study were in the same range as the literature. Halamickova and Malota [37] measured the level of TBARS in marinated Atlantic herring muscles collected in a market. The lowest TBARS levels were in warm baked marinades (1.17 ± 0.40 mg MDA/kg muscle) while the highest TBARS levels were observed in warm cooked marinades (16.48 ± 4.22 mg MDA/kg muscle). The MDA levels of Baltic herring fillets in our study after four months of storage varied from 1.0 ± 0.1 μM/g dm–2.1 ± 0.2 μM/g dm corresponding to approximately 4–8 mg MDA/kg muscle, which is the same level as the warm baked marinated Atlantic herring [37].

Baltic herring is rich in PUFAs which are susceptible to oxidation. The risk of oxidation is further increased due to the high content of pro-oxidative heme proteins found in the muscle of this species [38]. The results of our study indicate that all the acids tested are able to maintain the commercial quality of Baltic herring fillets for four months in terms of oxidation. However, the differences between the acids are significant, highlighting the need for careful selection of the acid to be used in preservation. The possible pro-oxidative effect of citric acid requires further studies.

Pickling and marinating with different weak acids resulted in different products in terms of the sensorial quality. The samples pickled and marinated with citric, lactic, malic, and tartaric acid did not have any differences in the intensities of the sensory attribute at the beginning of the storage, whereas the acetic acid sample was noticeably different from the other samples in nine sensorial properties. These were total intensity of odor and taste, intensity of vinegary odor and flavor, crumbly and moist texture, sour taste, fishy flavor, and metallic flavor. Later during the storage, the same attributes, apart from moist texture, continued to demonstrate differences between the samples. 

The sensory attributes indicated that the strong and distinctive vinegary character of acetic acid dominated in the sample pickled and marinated in acetic acid. The strong vinegary odor and flavor as well as the sourness identified in this sample may have suppressed other sensory properties of the pickled fish. The dominating sensory attributes in the other samples were the total intensity of odor and flavor, a fishy odor and a flavor and sweet taste, as well as a crumbly and moist texture. However, the intensities of these attributes were not strong. These results clearly indicate that by replacing acetic acid with other weak acids frequently used in food industry, pickled and marinated fish products that have novel and milder sensory profiles can be produced.

In the sensory analysis, the crumbliness of the texture of the samples increased. The intensity of the crumbly texture was already significantly higher in the acetic acid treated sample at the beginning of the storage compared to the other samples. The intensity of crumbliness increased throughout the study in all the samples but at different rates: in acetic, citric, or lactic acid treated samples the crumbliness increased steadily throughout the study, whereas in the sample treated with malic acid the significant increase took place only between 1 and 4 months and in the sample treated with tartaric acid between 0 and 1 month. The development of crumbliness was most likely due to the changes in the protein structure and solubility. There was also a trend for the moist texture in all the samples to increase in intensity, although not all of them showed a significant increase. This, together with the chemical analysis, indicates that during marination several processes caused by the pH and the salt content affect the moisture of the meat. However, the current data is not sufficient enough to identify a causal relationship between chemical analysis and sensory data.

No statistically significant indication on the development of a rancid odor and flavor was detected in the sensory evaluations despite the chemically measured increase of MDA in the samples. This was the case especially in the citric acid sample. Refsgaard et al. [39] found that the metallic flavor of cooked salmon correlated with hexanal, a main oxidation product of linoleic acid. The sample pickled and marinated in citric acid showed a statistically significant increase in the intensity the metallic flavor. However, the intensity of the metallic flavor also increased in all the other samples except the sample pickled and marinated in malic acid, although it also showed oxidation over time. The sample pickled and marinated in tartaric acid did not show any oxidation over time. Therefore, it is likely that the levels of MDA formation measured by chemical analyses are not reflected in the sensory quality.

Sensory profile studies of pickled or marinated herring are very limited. However, Nielsen et al. [26,27] studied the sensory properties of Atlantic and Baltic herring pickled in acetic acid. The sensory attributes of the profiles are similar to those of this study. The attributes in the studies described the sweetness and saltiness of the marinade, the sour and vinegary properties of acetic acid, as well as the fishy and metallic odor and flavor of herring. Rancidity was also included in the sensory attributes. Texture properties were also evaluated with the following attributes: firmness, elasticity, fatty mouth feel, juicy, and gritty. The texture properties in this study were evaluated with only two attributes describing the fracturability and moistness of the samples. However, the previous studies also included samples with Atlantic herring which contains more fat and can be larger in size than Baltic herring. This affects the descriptors in the sensory profile. Furthermore, the pickling process was different between this and the previous studies; this study utilized a considerably lower level of salt which was most likely the reason for the crumbly texture.

## 5. Conclusions

In conclusion, pickling with citric, lactic, malic, and tartaric acids in addition to acetic acid with pH levels of 3.7–4.2 resulted in pickled Baltic herring products with high microbiological quality similar to the traditional pickling with acetic acid. The results of the chemical analysis of the samples indicate that pickling and storage in marinade influenced the chemical composition of fish, e.g., by increasing the moisture and decreasing the protein content of the samples. Fat content decreased during the storage period in acetic acid and malic acid samples, and minor lipid oxidation took place especially in the citric acid sample. The flavor profiles of the samples other than the acetic acid sample were mild, especially in sourness and vinegariness. The fishy odor and flavor were perceived more strongly in citric, lactic, malic, and tartaric acid samples compared to the acetic acid sample. Overall, the results indicate that Baltic herring pickled and marinated in other food grade organic acids than the traditionally used acetic acid resulted in preservable fish products with novel and milder sensory profiles.

## Figures and Tables

**Figure 1 foods-11-01717-f001:**
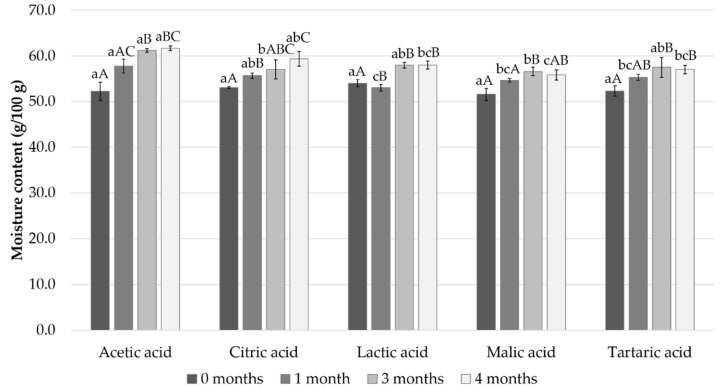
Measured moisture content after 0, 1, 3, and 4 months of storage. Different small letters within the same month and capital letters within the same acid indicate a statistically significant difference on a 95% confidence level.

**Figure 2 foods-11-01717-f002:**
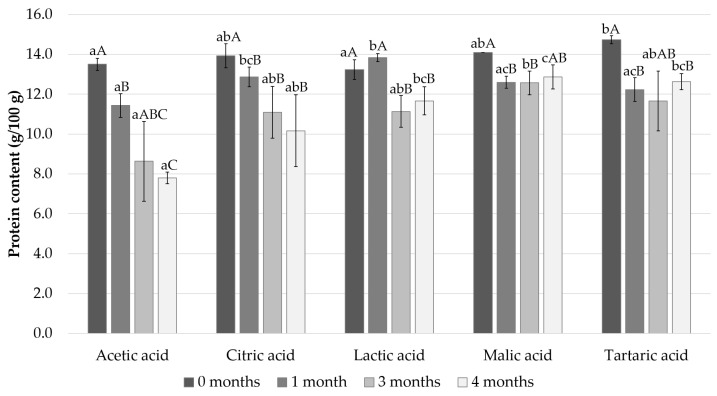
Measured protein content after 0, 1, 3, and 4 months of storage. Different small letters within the same month and capital letters within the same acid indicate a statistically significant difference on a 95% confidence level.

**Figure 3 foods-11-01717-f003:**
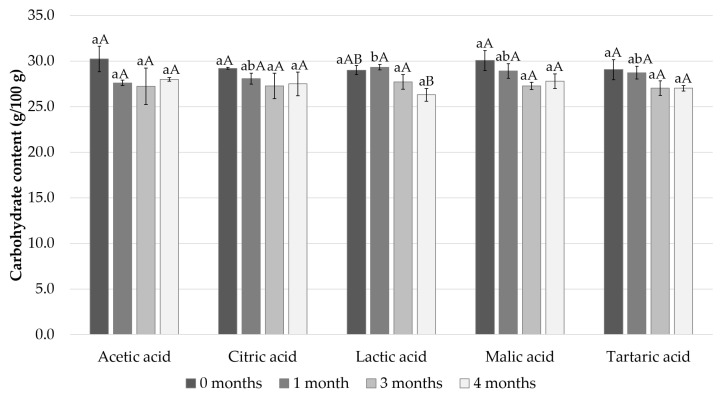
Calculated carbohydrate content after 0, 1, 3, and 4 months of storage. Different small letters within the same month and capital letters within the same acid indicate a statistically significant difference on a 95% confidence level.

**Figure 4 foods-11-01717-f004:**
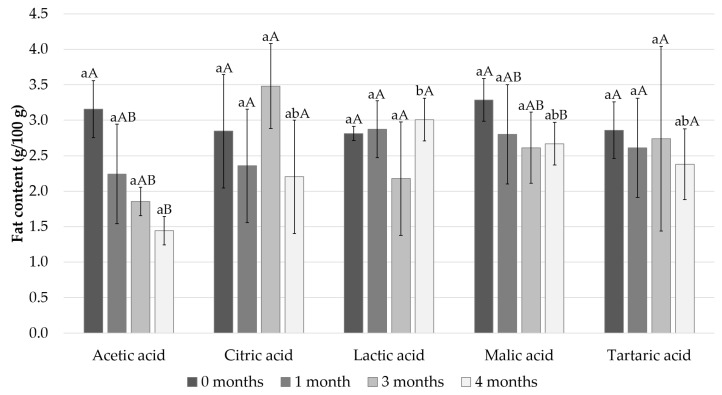
Measured fat content after 0, 1, 3, and 4 months of storage. Different small letters within the same month and capital letters within the same acid indicate a statistically significant difference on a 95% confidence level.

**Figure 5 foods-11-01717-f005:**
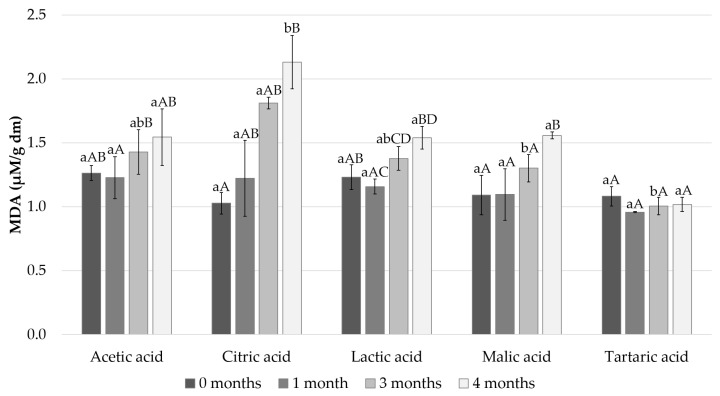
MDA content of the pickled and marinated Baltic herring fillets measured after 0, 1, 3 and 4 months in cold storage. Different small letters within the same month and capital letters within the same acid indicate a statistically significant difference on a 95% confidence level. Dm = dry matter.

**Table 1 foods-11-01717-t001:** Concentration and pH values of the pickling and marinade solutions.

Acid	Concentration in Pickling Solution (g/L)	Concentration in Marinade Solution (g/L)	pH of Pickling Solution	pH at the End of Pickling
Acetic acid	50.0	0.67	2.40 ± 0.00	3.85 ± 0.01
Citric acid	20.0	0.17	1.90 ± 0.01	3.63 ± 0.01
Lactic acid	20.0	0.16	2.10 ± 0.01	3.56 ± 0.01
Malic acid	20.0	0.14	2.03 ± 0.01	3.53 ± 0.01
Tartaric acid	21.5	0.11	1.78 ± 0.01	3.24 ± 0.01

**Table 2 foods-11-01717-t002:** The attributes and reference samples used in the sensory profiling. The intensities on a scale from 0–10 (0 = not at all, 10 = very strong) of the reference samples are in parenthesis.

Sensory Properties	Attribute	Reference Sample (Intensity)
Odor	Total intensity	Verbal instructions
	Sweet	Felix American pickled cucumber (5)
	Vinegar	1.25% (*v*/*v*) acetic acid (5)
	Fish	Verbal instructions
	Stuffy	Verbal instructions
	Rancid	Verbal instructions
	Metallic ^1^	Verbal instructions
Texture	Crumbly	Pirkka salad cheese, lactose free (9)
	Moistness	Pirkka salad cheese, lactose free (3)
Taste	Total intensity	Verbal instructions
	Sweet	2.0% (*w*/*v*) sucrose (5)
	Salty	0.2% (*w*/*v*) NaCl (5)
	Umami	0.1% and 0.2% (*w*/*v*) L-Glutamic acid monosodium salt monohydrate (5 and 8, respectively) ^2^
	Sour	0.04% (*w*/*v*) citric acid (5)
	Bitter	0.04% (*w*/*v*) caffeine (6)
Flavor	Vinegar	0.63% (*v*/*v*) acetic acid (6)
	Fish	Verbal instructions
	Stuffy	Verbal instructions
	Rancid	Verbal instructions
	Metallic	Verbal instructions

^1^ Metallic odor was evaluated at 1 and 4 months. ^2^ 0.1% (*w*/*v*) L-Glutamic acid monosodium salt monohydrate was included as a reference sample of umami taste 1- and 4-month’s evaluations.

**Table 3 foods-11-01717-t003:** Aerobic microorganisms (cfu/g log_10_) in the pickled and marinated fish fillet samples determined at 30 °C after 0, 1, 3, and 4 months of storage.

Acid	0 Months	1 Month	3 Months	4 Months
Acetic acid	1.00	<1.00	n.d. ^1^	<1.00
Citric acid	2.58	1.00	n.d. ^1^	1.00
Lactic acid	2.01	1.40	n.d. ^1^	1.40
Malic acid	1.85	1.30	n.d. ^1^	1.30
Tartaric acid	1.48	<1.00	n.d. ^1^	<1.00

^1^ not determined.

**Table 4 foods-11-01717-t004:** Measured pH values after 0, 1, 3, and 4 months of storage.

Acid	0 Months	1 Month	3 Months	4 Months
Acetic acid	3.78 ± 0.06 ^abA^	3.83 ± 0.03 ^abA^	3.91 ± 0.03 ^abA^	3.80 ± 0.08 ^Aa^
Citric acid	4.02 ± 0.04 ^Ca^	4.13 ± 0.05 ^Cb^	4.14 ± 0.08 ^Cb^	4.19 ± 0.12 ^Cab^
Lactic acid	3.96 ± 0.02 ^Ca^	3.98 ± 0.06 ^bcA^	4.12 ± 0.04 ^Cb^	4.09 ± 0.11 ^bcAB^
Malic acid	3.87 ± 0.01 ^Ba^	3.86 ± 0.09 ^abA^	4.02 ± 0.05 ^bcB^	3.93 ± 0.04 ^abAB^
Tartaric acid	3.68 ± 0.09 ^Aa^	3.79 ± 0.17 ^Aab^	3.92 ± 0.18 ^Ab^	3.87 ± 0.12 ^Ab^

^a–c^ Different letter indicates statistically significant difference between samples at each time point’s evaluation. ^A–C^ Different letter within an attribute indicates statistically significant difference between each time point’s evaluation of each acid sample.

**Table 5 foods-11-01717-t005:** The means and standard deviations of sensory attributes of Baltic herring pickled and marinated in different weak acids after 0, 1, and 4 months of storage evaluated on a scale from 0–10 (0 = not at all, 10 = very strong).

Attribute ^1^	Months	Acetic Acid	Citric Acid	Lactic Acid	Malic Acid	Tartaric Acid
O_total	0	7.4 ± 1.8 ^aA^	4.6 ± 1.8 ^bA^	5.4 ± 1.5 ^bA^	4.8 ± 1.5 ^bA^	5.7 ± 1.7 ^bA^
	1	8.4 ± 0.9 ^aB^	4.7 ± 1.5 ^bA^	5.5 ± 1.6 ^bcA^	5.4 ± 1.1 ^bcA^	6.0 ± 1.5 ^cA^
	4	8.1 ± 1.4 ^aAB^	4.9 ± 1.4 ^bA^	5.5 ± 1.7 ^bA^	5.0 ± 1.6 ^bA^	5.7 ± 1.3 ^bA^
O_sweet	0	2.4 ± 1.5 ^aA^	1.9 ± 1.5 ^aA^	1.7 ± 1.2 ^aA^	1.6 ± 1.5 ^aA^	1.8 ± 1.5 ^aA^
	1	2.3 ± 1.2 ^aA^	1.9 ± 1.3 ^aA^	2.0 ± 1.6 ^aA^	2.1 ± 1.6 ^aA^	1.9 ± 1.8 ^aA^
	4	2.9 ± 2.2 ^aA^	2.2 ± 1.4 ^aA^	2.4 ± 1.7 ^aA^	2.1 ± 1.7 ^aA^	2.7 ± 1.5 ^aA^
O_vinegar	0	6.7 ± 2.8 ^aA^	2.0 ± 2.0 ^bA^	2.0 ± 2.4 ^bA^	1.9 ± 1.9 ^bA^	1.9 ± 1.9 ^bA^
	1	8.1 ± 1.2 ^aB^	1.6 ± 1.3 ^bA^	2.0 ± 1.7 ^bA^	2.4 ± 1.8 ^bA^	2.5 ± 2.2 ^bA^
	4	8.6 ± 1.0 ^aAB^	3.0 ± 2.1 ^bA^	3.7 ± 2.3 ^bA^	2.7 ± 2.2 ^bA^	3.2 ± 2.2 ^bA^
O_fish	0	3.1 ± 2.0 ^aA^	3.4 ± 2.2 ^aA^	4.2 ± 2.2 ^aA^	4.3 ± 2.2 ^aA^	4.6 ± 2.5 ^aA^
	1	2.7 ± 1.4 ^aA^	3.5 ± 1.9 ^abA^	4.3 ± 1.6 ^bA^	4.0 ± 1.5 ^bA^	4.3 ± 1.5 ^bA^
	4	2.5 ± 1.6 ^aA^	4.1 ± 1.7 ^bA^	4.8 ± 1.8 ^bA^	4.9 ± 1.6 ^bA^	5.0 ± 1.6 ^bA^
O_stuffy	0	1.3 ± 1.8 ^aA^	2.0 ± 2.3 ^aA^	1.7 ± 2.3 ^aA^	1.7 ± 2.0 ^aA^	2.0 ± 2.2 ^aA^
	1	0.4 ± 0.6 ^aA^	2.1 ± 1.9 ^bA^	1.7 ± 1.5 ^bA^	1.2 ± 1.1 ^bA^	1.8 ± 1.6 ^bA^
	4	1.2 ± 1.5 ^aA^	2.1 ± 1.7 ^aA^	2.2 ± 1.8 ^aA^	1.8 ± 1.5 ^aA^	2.0 ± 1.3 ^aA^
O_rancid	0	0.9 ± 2.1 ^aA^	1.7 ± 2.4 ^aA^	1.7 ± 2.6 ^aA^	2.1 ± 2.6 ^aA^	2.2 ± 2.9 ^aA^
	1	1.2 ± 2.3 ^aA^	2.3 ± 2.5 ^aA^	2.4 ± 2.4 ^aA^	2.0 ± 2.2 ^aA^	2.3 ± 2.3 ^aA^
	4	0.5 ± 1.1 ^aA^	2.0 ± 2.5 ^aA^	1.7 ± 1.7 ^aA^	1.7 ± 1.9 ^aA^	2.0 ± 2.2 ^aA^
O_metallic	1	0.8 ± 0.9 ^aA^	1.3 ± 1.2 ^aA^	1.1 ± 1.2 ^aA^	1.6 ± 1.1 ^aA^	1.8 ± 1.5 ^aA^
	4	1.6 ± 1.8 ^aA^	1.9 ± 1.7 ^aA^	2.2 ± 2.0 ^aB^	2.1 ± 1.9 ^aA^	2.0 ± 1.1 ^aA^
Tx_crumbly	0	8.0 ± 1.9 ^aA^	5.4 ± 1.9 ^bA^	6.2 ± 1.9 ^bA^	6.3 ± 1.8 ^bA^	6.4 ± 1.8 ^bA^
	1	9.4 ± 0.9 ^aB^	7.5 ± 1.7 ^bcB^	7.5 ± 1.6 ^bcB^	7.2 ± 1.6 ^bA^	8.4 ± 1.1 ^cB^
	4	9.9 ± 0.3 ^aC^	9.3 ± 0.7 ^bC^	9.4 ± 1.1 ^abC^	8.8 ± 1.2 ^bB^	9.2 ± 1.1 ^abB^
Tx_moistness	0	7.0 ± 1.3 ^aA^	5.4 ± 1.4 ^bA^	5.1 ± 1.4 ^bA^	5.7 ± 1.4 ^bA^	5.4 ± 1.5 ^bA^
	1	7.3 ± 1.8 ^aA^	5.8 ± 1.5 ^bA^	5.5 ± 1.5 ^bAB^	5.8 ± 1.1 ^bA^	6.4 ± 1.7 ^abB^
	4	7.7 ± 2.0 ^aA^	6.6 ± 2.0 ^aA^	6.6 ± 1.9 ^aB^	6.3 ± 2.0 ^aA^	6.2 ± 1.9 ^aAB^
T_total	0	8.7 ± 0.5 ^aA^	5.7 ± 1.2 ^bA^	5.7 ± 1.1 ^bA^	6.1 ± 0.9 ^bA^	6.2 ± 1.0 ^bA^
	1	9.0 ± 0.5 ^aAB^	5.8 ± 1.0 ^bA^	6.0 ± 1.1 ^bA^	5.6 ± 1.2 ^bA^	6.2 ±1.4 ^bAB^
	4	9.3 ± 0.7 ^aB^	6.6 ± 1.2 ^bB^	6.5 ± 1.2 ^bA^	6.0 ± 1.6 ^bA^	7.0 ± 1.3 ^bB^
T_sweet	0	4.1 ± 2.3 ^aA^	5.2 ± 2.1 ^aA^	5.4 ± 2.0 ^aA^	5.1 ± 1.9 ^aA^	5.4 ± 2.3 ^aA^
	1	3.2 ± 2.1 ^aA^	5.0 ± 2.2 ^aA^	4.4 ± 2.2 ^aA^	4.9 ± 2.1 ^aA^	4.9 ± 2.4 ^aA^
	4	4.7 ± 2.3 ^aA^	6.2 ± 2.1 ^aA^	6.2 ± 2.0 ^aA^	6.1 ± 2.0 ^aA^	6.1 ± 2.4 ^aA^
T_salt	0	1.9 ± 1.7 ^aA^	1.9 ± 1.5 ^aA^	1.9 ± 1.9 ^aAB^	2.0 ± 1.6 ^aA^	2.0 ± 1.7 ^aA^
	1	1.9 ± 2.1 ^aA^	1.6 ± 1.5 ^aA^	1.5 ± 1.5 ^aA^	1.9 ± 1.9 ^aA^	1.8 ± 1.9 ^aA^
	4	2.5 ± 2.6 ^aA^	3.0 ± 2.5 ^aA^	3.1 ± 2.3 ^aB^	3.0 ± 2.3 ^aA^	3.1 ± 2.3 ^aA^
T_umami	0	0.8 ± 1.3 ^aA^	1.4 ± 1.6 ^aA^	1.4 ± 1.6 ^aA^	1.4 ± 1.5 ^aA^	1.1 ± 1.3 ^aA^
	1	1.0 ± 1.4 ^aA^	1.2 ± 1.4 ^aA^	1.5 ± 1.7 ^aA^	1.3 ± 1.3 ^aA^	1.8 ± 1.8 ^aA^
	4	0.7 ± 1.0 ^aA^	1.0 ± 1.1 ^aA^	1.4 ± 1.3 ^aA^	1.3 ± 1.4 ^aA^	1.1 ± 1.2 ^aA^
T_sour	0	6.3 ± 1.3 ^aA^	3.0 ±1.3 ^bAB^	2.9 ± 1.3 ^bA^	3.2 ± 1.6 ^bA^	3.6 ± 1.5 ^bA^
	1	6.9 ± 1.5 ^aA^	2.7 ± 1.5 ^bA^	2.8 ± 1.4 ^bA^	3.0 ± 1.2 ^bA^	3.5 ± 1.3 ^bA^
	4	7.6 ± 1.1 ^aB^	4.1 ± 2.1 ^bB^	4.0 ± 2.1 ^bA^	4.2 ± 2.1 ^bA^	5.3 ± 2.1 ^bB^
T_bitter	0	1.2 ± 1.3 ^aA^	0.9 ± 2.1 ^aA^	0.8 ± 1.9 ^aA^	0.9 ± 1.9 ^aA^	1.2 ± 1.9 ^aA^
	1	1.8 ± 0.9 ^aA^	1.6 ± 2.1 ^aA^	2.0 ± 1.5 ^aAB^	1.1 ± 1.7 ^aA^	1.8 ± 2.1 ^aA^
	4	1.5 ± 1.0 ^aA^	1.9 ± 2.2 ^aA^	2.6 ± 2.2 ^aB^	1.7 ± 1.9 ^aA^	2.5 ± 1.9 ^aA^
F_vinegar	0	8.0 ± 1.1 ^aA^	3.4 ± 1.1 ^bA^	2.7 ± 1.5 ^bAB^	3.9 ± 1.3 ^bA^	3.8 ± 1.5 ^bA^
	1	8.5 ± 1.7 ^aA^	2.9 ± 1.6 ^bA^	2.3 ± 1.8 ^bA^	3.3 ± 1.1 ^bA^	3.7 ± 1.5 ^bA^
	4	9.0 ± 1.5 ^aA^	3.9 ± 2.2 ^bA^	4.2 ± 2.5 ^bB^	3.6 ± 1.4 ^bA^	5.1 ± 2.1 ^bA^
F_fish	0	3.2 ± 1.5 ^aA^	4.6 ± 1.7 ^bA^	4.7 ± 1.7 ^bA^	4.8 ± 1.8 ^bA^	4.8 ± 2.0 ^bA^
	1	2.8 ± 1.4 ^aA^	4.2 ± 1.6 ^bA^	4.2 ± 1.7 ^bA^	4.3 ± 1.5 ^bA^	4.3 ± 1.8 ^bA^
	4	2.9 ± 1.4 ^aA^	4.8 ± 1.3 ^bA^	5.1 ± 1.6 ^bA^	5.0 ± 1.3 ^bA^	4.8 ± 1.4 ^bA^
F_stuffy	0	0.6 ± 1.3 ^aA^	1.0 ± 1.5 ^aA^	0.9 ± 1.6 ^aA^	1.4 ± 2.2 ^aA^	1.1 ± 1.4 ^aA^
	1	0.7 ± 1.2 ^aA^	1.2 ± 1.8 ^aA^	1.5 ± 1.2 ^aA^	1.1 ± 1.8 ^aA^	1.7 ± 2.4 ^aAB^
	4	0.7 ± 1.2 ^aA^	1.8 ± 1.8 ^abA^	2.1 ± 2.0 ^abA^	1.4 ± 1.4 ^abA^	2.7 ± 2.1 ^bB^
F_rancid	0	0.5 ± 1.5 ^aA^	1.6 ± 2.3 ^aA^	1.7 ± 2.5 ^aA^	1.2 ± 2.1 ^aA^	2.1 ± 3.0 ^aA^
	1	1.2 ± 2.0 ^aA^	1.5 ± 2.0 ^aA^	1.9 ± 2.2 ^aA^	1.9 ± 2.2 ^aA^	2.1 ± 2.2 ^aA^
	4	1.1 ± 1.3 ^aA^	1.7 ± 1.8 ^aA^	2.5 ± 2.0 ^aA^	1.5 ± 1.8 ^aA^	2.6 ± 2.2 ^aA^
F_metallic	0	1.0 ± 1.3 ^aA^	1.4 ± 1.4 ^abA^	1.9 ± 1.9 ^abA^	2.1 ± 2.0 ^abA^	2.4 ± 1.8 ^bA^
	1	1.4 ± 1.2 ^aB^	1.9 ± 1.5 ^abA^	2.7 ± 1.8 ^bA^	1.9 ± 1.6 ^abA^	2.7 ± 1.6 ^bAB^
	4	2.7 ± 1.9 ^abB^	2.9 ± 1.8 ^abB^	3.8 ± 1.8 ^abB^	2.6 ± 1.8 ^aA^	4.1 ± 1.8 ^bB^

^1^ O = odor, Tx = texture, T = taste, F = flavor. ^a–c^ Different letter indicates statistically significant difference between samples at each time point’s evaluation. ^A–C^ Different letter within an attribute indicates statistically significant difference between each time point’s evaluation of each acid sample.

## Data Availability

The data presented in this study are available on request from the corresponding author. The data are not publicly available due to privacy.

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
