# Peer review of "Effects of Weak Acids on the Microbiological, Nutritional and Sensory Quality of Baltic Herring (*Clupea harengus membras*)"

_foods, 2022, doi:10.3390/foods11121717_

Round 1
Reviewer 1 Report
This manuscript explored the influence of acetic, citric, lactic, malic, and tartaric acids on the preservability and quality of pickled and marinated Baltic herring was studied by measuring microbiological quality, pH, chemical composition, and lipid oxidation and by sensory profiling. This study evaluated the potential of weak acids to produce safe and nutritious pickled fish products and provide some data support for other acids to replace acetic acid pickling. The structure of the article is basically complete and the grammar is correct, but there still have many problems. Therefore, I suggest that the author make a major revision or supplement. The detail suggestion will be as followed:
- Please supplement the pH value, moisture content, size of fish fillets and other information of raw materials.
- Are the acids used in the experiments food-grade or chemical analysis grade?
- What is the basis for determining the ratio of various acids in this study? Why is the initial and final pH of the pickling solution different with the addition of various acids?
- Why are the various acids added in different amounts? How to verify whether the difference in test effect is caused by different acid types or different ionic strengths?
- The quality of pickled products is an important criterion for testing the pickling effect. However, the content of this study is insufficient in terms of texture, colour, moisture, flavour, etc., and it is recommended to supplement them appropriately.
- The results in "composition" show that the water content, protein content, carbohydrate content and fat content in 0 months of each treatment group are different. This indicates that there are differences in all treatment groups of raw materials selected in this study. Therefore, is the comparison of differences between groups conducted later reliable?
- Why did the protein content decrease and then increase during the pickling process of the Lactic acid, Malic acid and Tartaric acid group? Please add the author's explanation.
- In the analysis of significance, p<0.05 is sufficient to indicate a significant difference.
- L284: “United States” to “The United States”.
- L291: “correction” to “corrections”.
- Please add a conclusion at the end of the article.
Reviewer 2 Report
The paper is well written and interesting, but in my opinion it would have been useful to measure biogenic amines and in particular histamine.
Reviewer 3 Report
The following issues should be handled.
1. The Editorial issues and typographical issues should be corrected throughout the document.
2. The abstract should be fine tune align with the results.
3. In-Line number 38, remove the "(p. 634)"
4. Line number 53, correct the sentence which is ending with the reference [4].
5. Line number 57, "in H+", The + should be superscript.
6. Lines from 76-78, The sentence has to revise once.
7. Line 81, remove " (p. 989)".
8. The Introduction is a little more detailed, in some time it looks like the discussion. For instance, Line number 105 to 108 are so particular. So, I suggest the introduction section should revise.
9. In the last paragraph (120,121), the Introduction is given as the aim and specifically the aim, and again in 124 given aim. So, I suggest you write this into a clear idea.
10. Line 153-154 give the specifications like size etc..
11. Line 247, given as the "professional panel". Is this a trained panel? or certified?
12. In all the tables after the heading, some information is provided. That information should be in the footnote.
13. All the tables have to check for the formating
14. Line 275- 279, is repeated. Check once.
15. Line 297- 298, the given lines are like the methodology. As this section is results.
16. In section 4, the discussion needs to check once with the previous studies and the standards etc.
17. Line 519 and 520 rewrite once.
18. Conclusion section is missing in the manuscript.
19. Check for the English throughout the manuscript for fine-tuning
Round 2
Reviewer 1 Report
The revised manuscript is OK now. Please accept it for publication in the present form.